# Assessing B-Z DNA Transitions in Solutions via Infrared Spectroscopy

**DOI:** 10.3390/biom13060964

**Published:** 2023-06-08

**Authors:** Mengmeng Duan, Yalin Li, Fengqiu Zhang, Qing Huang

**Affiliations:** 1Henan Key Laboratory of Ion-Beam Bioengineering, School of Physics and Microelectronics, Zhengzhou University, Zhengzhou 450052, China; 202012132012347@gs.zzu.edu.cn; 2School of Food and Biological Engineering, Henan University of Animal Husbandry and Economy, Zhengzhou 450047, China; 201087@hnuahe.edu.cn; 3CAS Key Laboratory of High Magnetic Field and Ion Beam Physical Biology, Institute of Intelligent Machines, Hefei Institutes of Physical Sciences, Chinese Academy of Sciences, Hefei 230031, China; 4Science Island Branch of Graduate School, University of Science and Technology, Hefei 230026, China

**Keywords:** Z-DNA, FTIR, CD

## Abstract

Z-DNA refers to the left-handed double-helix DNA that has attracted much attention because of its association with some specific biological functions. However, because of its low content and unstable conformation, Z-DNA is normally difficult to observe or identify. Up to now, there has been a lack of unified or standard analytical methods among diverse techniques for probing Z-DNA and its transformation conveniently. In this work, NaCl, MgCl_2_, and ethanol were utilized to induce d(GC)_8_ from B-DNA to Z-DNA in vitro, and Fourier transform infrared (FTIR) spectroscopy was employed to monitor the transformation of Z-DNA under different induction conditions. The structural changes during the transformation process were carefully examined, and the DNA chirality alterations were validated by the circular dichroism (CD) measurements. The Z-DNA characteristic signals in the 1450 cm^−1^–900 cm^−1^ region of the d(GC)_8_ infrared (IR) spectrum were observed, which include the peaks at 1320 cm^−1^, 1125 cm^−1^ and 925 cm^−1^, respectively. The intensity ratios of A_1320_/A_970_, A_1125_/A_970_, and A_925_/A_970_ increased with Z-DNA content in the transition process. Furthermore, compared with the CD spectra, the IR spectra showed higher sensitivity to Z-DNA, providing more information about the molecular structure change of DNA. Therefore, this study has established a more reliable FTIR analytical approach to assess BZ DNA conformational changes in solutions, which may help the understanding of the Z-DNA transition mechanism and promote the study of Z-DNA functions in biological systems.

## 1. Introduction

Deoxyribonucleic acid (DNA), as the hereditary material in biological systems, plays an important role in life. Structural changes of DNA molecules affect the expression of genes inside the nuclei of cells. Unlike Watson–Crick’s B-DNA, Z-DNA is a left-handed double-stranded helix with a zigzag shape that can be transformed from B-DNA [1]. It was found that GC bases are more easily induced to Z-DNA [2,3,4,5], and some base modifications such as methylation of cytosine and guanine and deamination of adenine may increase the stability of Z-DNA [6]. At present, a variety of biological functions of Z-DNA have been revealed. In general, it is involved in gene regulation [7,8]. For example, it may induce genetic instability and cause a variety of diseases [2,9,10,11]. Evidence has shown that Z-DNA appears in the hippocampus of patients with Alzheimer’s disease, and it is speculated that Z-DNA may be related to neurodegenerative diseases [12]. However, currently, there are very few reports on Z-DNA research, not only because of the lack of effective tools for Z-DNA detection but also due to its low content existing in nature.

In order to study the properties of Z-DNA, researchers have induced the formation of Z-DNA in vitro under special conditions such as high ionic strength, chemical modification, and the use of binding proteins and negative superhelices [13,14,15,16,17,18,19,20,21,22,23]. For the detection and analysis of Z-DNA, spectroscopic methods have been proposed and employed, such as X-ray diffraction, circular dichroism (CD), and nuclear magnetic resonance (NMR) spectroscopy [1,13,21,24,25,26,27,28,29]. However, these methods either require expensive instruments and complicated operation or cannot assess the B-Z DNA transition instantly and quantitatively.

Fourier transform infrared (FTIR) spectroscopy can probe the biochemical and structural information of analyte samples. When the composition and/or conformation of DNA change, the peak frequency and intensity of the infrared spectroscopy will change correspondingly. Currently, FTIR spectroscopy has already been applied in DNA structure analysis [30,31]. The application of FTIR spectroscopy in the study of DNA conformation has also gained encouraging attention [8,32,33,34,35,36]. However, previous research on Z-DNA through FTIR had some limitations, such as the requirement of low hydration to achieve the purpose of Z-DNA detection [30,37,38]. This hydration method may lead to changes in the liquid environment and so bring unpredictable effects on the formation of Z-DNA. On the other hand, it is still difficult to conduct a quantitative analysis of Z-DNA because of insufficient identification of the Z-DNA characteristic IR bands [39,40].

In order to establish a more reliable approach for Z-DNA assessment, we tried to induce the transformation of B-DNA into Z-DNA in solutions by a variety of standard inducers (NaCl, MgCl_2_, and ethanol) without reduction of relative humidity, and then applied FTIR spectroscopy to detect and analyze the gradual change process of the B-Z DNA transition. It is known that NaCl, MgCl_2_, and ethanol are Z-DNA inducers with good repeatability [20,21,41,42,43]. Especially DNA with multiple GC has a strong tendency to form Z-DNA under these conditions [2,3,4]. Therefore, in the experiment, different concentrations of NaCl, MgCl_2_, and ethanol solutions were used to induce the dsDNA of d(GC)_8_ from B-DNA to Z-DNA. The Z-DNA transformation under different conditions was thus observed and evaluated; in particular, the IR band at 970 cm^−1^ was used as the internal standard for the potential quantitative analysis, and based on the evaluation of the intensity ratios (A_1320_/A_970_, A_1125_/A_970_, and A_925_/A_970_), the B-Z DNA transition process under some specific conditions was inspected and assessed.

## 2. Materials and Methods

### 2.1. Materials

The d(GC)_8_ sequence used in the experiment was purchased from General Biology (Hefei, China). The concentration was 5 OD/tube (the OD value was measured at 260 nm in the ultraviolet absorption spectrum) and the MW was 4885.16 g/mole, equivalent to 181.6 μg/tube. Tm (the temperature at which 50% nucleic acids become denatured [44,45]) was 63.9 °C. The solution was dissolved in buffer (50 mM NaCl, 5 mM Tris-HCl, pH = 8.0), and the A_260_/A_280_ = 1.8–2.2 was determined by UV spectrophotometer, confirming that the DNA had high purity. Ultra-pure Milli-Q water was used in all the experiments. The NaCl (analytically pure) used in the experiment was purchased from Tianjin Hengxing Chemical Reagent Manufacturing Co., Ltd. (Tianjin, China), and the MgCl_2_ (analytically pure) was purchased from Tianjin Zhiyuan Chemical Reagent Co., Ltd. (Tianjin, China). Absolute ethanol was purchased from Beijing Huateng Chemical Co., Ltd. (Beijing, China), and the Tris (analytically pure) was from Solarbio (Beijing, China).

### 2.2. Induction of Z-DNA

Firstly, the purchased ssDNA was combined into dsDNA using the base complementary pairing principle. The dry nucleic acid powder was first dissolved with buffer solution (50 mM NaCl, 5 mM Tris-HCl, pH = 8.0) and then incubated at 37 °C for 30 min to obtain stable dsDNA [41], used for CD and UV detection. NaCl, MgCl_2_, and ethanol were prepared in buffer solutions (50 mM NaCl, 5 mM Tris-HCl, pH = 8.0). The prepared inducer solution was added to the dsDNA solution. The final concentrations of NaCl were 1 M, 2 M, 3 M, 4 M, 4.5 M, and 5 M, respectively; the final concentrations of MgCl_2_ were 0.5 M, 1 M, 1.5 M, 2 M, 2.5 M, and 3 M, respectively; and the volume ratios of ethanol were 0.4, 0.5, 0.55, 0.6, 0.65, and 0.7, respectively. After adding the Z-DNA inducer solution, the DNA samples were incubated at 37 °C for more than 4 h to stabilize the structure of the dsDNA.

### 2.3. FTIR Spectral Analysis

Before the FTIR spectral measurements, the dry nucleic acid powder was dissolved with ultra-pure Milli-Q water and incubated at 37 °C for 30 min to obtain stable dsDNA. Then the dsDNA was added to a CaF_2_ window piece drop by drop and put into a drying oven keeping the temperature at 37 °C for about 20 min until the water of the DNA solution evaporated on the window piece. Then, 5 μL of different inducing solutions were added on the film formed by the dsDNA, the film was quickly covered with another clean CaF_2_ window piece, and the connection of the window piece was sealed with sealing film to reduce the evaporation of water. The final concentrations of NaCl were 50 mM, 1 M, 2 M, 3 M, 4 M, 4.5 M, and 5 M, respectively; the final concentrations of MgCl_2_ were 0 M, 0.5 M, 1 M, 1.5 M, 2 M, 2.5 M, and 3 M, respectively; and the volume ratios of ethanol were 0, 0.4, 0.5, 0.55, 0.6, 0.65, and 0.7, respectively. The concentration of the DNA solution was 36.32 μg/μL. The structural changes of the dsDNA were detected by a Thermo Fisher Scientific Nicolet iS10 infrared spectrometer.

The parameters of the infrared spectrometer were set as follows: the spectral range was from 4000 cm^−1^ to 900 cm^−1^, the spectral resolution was 4 cm^−1^, and the number of scans was 30. The software Origin 2021 was used for the spectral analysis. At least three replicates were set for each measurement.

### 2.4. Circular Dichroism Analysis

Circular dichroism was conducted using a Chirascan^TM^ Circular dichroism spectrometer. The spectral scanning range was 200 nm to 350 nm, with the scanning time being 0.25 s per point. The solution sample was put into a quartz colorimetric tube with a diameter of 1 mm. Each CD spectrum was scanned three times on average. Ultra-pure water was used as a blank control.

## 3. Results

Figure 1 shows the FTIR spectra of d(GC)_8_ dsDNA, with the spectral assignments given in Table 1. In this work, we mainly focused on the 1450 cm^−1^−900 cm^−1^ region of DNA infrared spectra. In addition, the peak at 970 cm^−1^ is attributed to the deoxyribose C-C stretching mode, and it is often used as an internal reference for normalization to analyze the structural changes of dsDNA [34,46].

### 3.1. NaCl-Induced Z-DNA

Firstly, the structural changes of dsDNA induced by different concentrations of NaCl were analyzed (Figure 2). The characteristic peaks at ~1125 cm^−1^ and ~925 cm^−1^ changed obviously with the increase of NaCl concentration, indicating the transition from B-DNA to Z-DNA. It was found that when the concentration of NaCl reached 3 M, the original 1068 cm^−1^ of FTIR spectrum disappeared, accompanied by the appearance of 1077 cm^−1^ and 1066 cm^−1^ (Figure 2a,b), and the CD spectrum at this concentration showed a trend of Z-DNA transition (Figure 2e). These results indicated that when NaCl was over 3 M, the transformation from B-DNA to Z-DNA occurred.

According to the peak positions indicated by the second derivative spectrum as shown in Figure 2b, the curve fitting of the infrared spectrum from 1150 cm^−1^ to 900 cm^−1^ was achieved (Figure 2c and Appendix A and Appendix A). Then, the intensity ratios of the Z-DNA characteristic peak area to the internal reference area (A_925_/A_970_ and A_1125_/A_970_) were used to evaluate the variation trend of Z-DNA content. The results showed that the relative Z-DNA content increased with the increase of NaCl concentration (Figure 2d), and the changes evaluated based on either A_925_/A_970_ or A_1125_/A_970_ were the same. It was found that substantial structural transformation of DNA occurred when the NaCl concentration was larger than 3 M. When the concentration reached 4 M, the ratios of A_925_/A_970_ and A_1125_/A_970_ reached a plateau, indicating that most B-DNA had been transformed into Z-DNA. In addition, I_925_/I_970_ and I_1125_/I_970_ changed with the same trend as A_925_/A_970_ and A_1125_/A_970_, both of which changed at 3 M significantly, and reached a plateau above 4 M (Appendix A).

The CD spectrum in the range of 220–350 nm reflects the interaction of base pairs in the helical structure [57]. In fact, Z-DNA has an obvious structural difference from B-DNA as shown by the CD measurement (Appendix A). According to the experimental results of d(GC)_8_, the CD of Z-DNA showed a positive value near 270 nm and a negative value near 295 nm, while the CD of B-DNA showed a negative value near 255 nm and a positive value near 285 nm [58]. BZ-DNA is the result of the coexistence of Z-DNA and B-DNA, which showed a negative band at 295 nm for Z-DNA and a negative value at 255 nm for B-DNA, and a positive peak between B-DNA and Z-DNA [59].

As shown by the CD spectra in Figure 2e, NaCl could induce the dsDNA transformation, which is consistent with previous studies [21,24,41,42,60]. B-DNA was obviously transformed to Z-DNA after treatment with NaCl with a concentration greater than 4 M, and its structural change was consistent with the concentration change trend of NaCl. Although the structure of dsDNA did not change significantly when the NaCl concentration was below 3 M, the peak intensity of the CD spectrum changed, indicating that the dsDNA conformation was varied.

In addition, as Pohl and Jovin found that the conversion of poly(dG-dC)·poly(dG-dC) from one conformation to another was associated with an increase in the A_295_/A_260_ ratio of the UV absorption spectrum, other research groups also applied this method to evaluate conversion [21,61,62,63]. Therefore, we used this for a double check. As shown in Appendix A, when NaCl reached 3 M, the UV absorption ratio A_295_/A_260_ gradually increased with the increase of NaCl concentration, further confirming that the Z-DNA structure was gradually transformed.

### 3.2. MgCl_2_-Induced Z-DNA

Furthermore, we applied the cation Mg^2+^ to treat d(GC)_8_ and analyzed the transformation law of Z-DNA. It was found that the effect of MgCl_2_ on dsDNA was similar to that caused by NaCl.

The analysis in Figure 3a,b shows that in the transformation from B-DNA to Z-DNA, the characteristic peaks at ~1125 cm^−1^ and ~925 cm^−1^ appeared, which are attributed to Z-DNA. When the concentration of MgCl_2_ reached 1.5 M, the peak at 1068 cm^−1^ disappeared, and the peaks at 1077 cm^−1^ and 1066 cm^−1^ appeared (Figure 3b), similar to the spectrum when the NaCl concentration reached 3 M. When MgCl_2_ increased to 2 M, the induction effect on dsDNA was further enhanced, and the FTIR spectrum showed new changes. Through curve fitting of the infrared spectrum from 1150 cm^−1^ to 900 cm^−1^ (Figure 3c and Appendix A with the fitting parameters given in Appendix A), we found that the ratio of the characteristic peak area to the internal reference area was closely related to the concentration of MgCl_2_ (Figure 3d), indicating the structural transformation of dsDNA. Combined with the CD spectrum and FTIR second derivative analysis, it was found that when the concentration of MgCl_2_ exceeded 3 M, the conformation of DNA gradually shifted into a new conformation. In addition to the area ratio of the characteristic peaks, we also analyzed the intensity ratios of the characteristic peaks (Appendix A), and found that the variation trend was the same as that of the area ratio.

As shown by the CD spectra in Figure 3e, MgCl_2_ could induce dsDNA transformation, which is consistent with previous studies [20,41,60]. Furthermore, the CD spectral analysis confirmed that the dsDNA structure was gradually transformed into Z-DNA when the concentration of MgCl_2_ reached more than 1.5 M (Figure 3e). Compared with NaCl, MgCl_2_ has more electric charge and has a stronger effect on Z-DNA. When the concentration of MgCl_2_ increased to 3 M, the CD spectrum of dsDNA was quite different from that of Z-DNA, indicating that the structure of DNA had changed again, which was consistent with previous reports [20,64]. Appendix A shows the UV absorption spectra and A_295_/A_260_ ratio after d(GC)_8_ induction. It shows that A_295_/A_260_ increased with the increase of MgCl_2_ concentration, confirming that an increase of MgCl_2_ would affect the structure of dsDNA.

### 3.3. Ethanol-Induced Z-DNA

In addition to the cationic inducers mentioned above, we also investigated Z-DNA induced by ethanol. DNA is insoluble in ethanol and soluble in an aqueous solution. It has been reported that the addition of ethanol could induce the formation of Z-DNA with a low hydration degree by reducing the hydration degree of nucleic acid [65]. In this case, we observed that the induction effect of ethanol on dsDNA conformation was different from that of NaCl and MgCl_2_. As shown in Figure 4a,b, the peaks at 1125 cm^−1^ and 925 cm^−1^ appeared when B-DNA changed to Z-DNA, and then the peaks for dsDNA furanose and phosphate groups at 1100 cm^−1^−1000 cm^−1^ changed. This is consistent with the change in MgCl_2_ treatment. However, when the volume ratio of ethanol reached 0.6, the peak at 1068 cm^−1^ disappeared in the second derivative spectrum, and the peaks at 1077 cm^−1^ and 1066 cm^−1^ appeared. When the volume ratio of ethanol ranged from 0.65 to 0.7, the FTIR spectrum of dsDNA changed obviously, and the peak at 1049 cm^−1^ disappeared, while the peaks at 1042 cm^−1^ and 1054 cm^−1^ appeared (Figure 4b). This change is different from the structural change in the case of MgCl_2_ treatment. To be noted, the structural changes of dsDNA induced by excessive ethanol were weaker than those induced by excessive MgCl_2_.

The relative change of Z-DNA was calculated by curve fitting in the range of 1150 cm^−1^–900 cm^−1^ of the infrared spectrum (Figure 4c and Appendix A with the fitting parameters given in Appendix A). It can be seen that the Z-DNA level is positively correlated with ethanol content (Figure 4d). When the volume ratio of ethanol reached 0.6, the ratio of Z-DNA increased. This implies that most B-DNA was converted to Z-DNA at the ethanol volume ratio of 0.6. In addition to the area ratio analysis, we also analyzed the intensity ratio between the characteristic peak and internal reference (Appendix A), and found that its variation trend was the same as that of the area ratio.

As shown by the CD spectra in Figure 4e, ethanol could induce dsDNA transformation, which is consistent with previous studies [42,43]. Furthermore, CD measurements confirmed that the dsDNA structure did change with the increase of ethanol content, and this change was correlated with the ethanol volume ratio (Figure 4e). For the unusual result with 0.4 proportion of ethanol, it was found that the characteristic peaks of B-DNA (i.e., 285 nm peaks) still remained, while the peak intensity at 255 nm was different from that of conventional B-DNA. Based on the literature [66], the special curve for the 0.4 proportion of ethanol might be due to the synergistic effect of ethanol and NaCl in the buffer solution. The ethanol promoted an interaction between NaCl and the anion in nucleic acid, resulting in the CD spectra of Z-DNA at 255 nm different from that of B-DNA. Johan H. van de Sande et al. reported the cationic promotion effect of ethanol and proved that, with an increase of ethanol volume and/or decrease of NaCl concentration, the induction effect of ethanol would be dominant [20]. It can also be seen from the CD spectrum that ethanol showed the strongest effect on dsDNA when the volume ratio of ethanol was 0.65, which is consistent with the changing trend based on the FTIR spectral analysis. Appendix A shows the changing trend of A_295_/A_260_ according to UV absorption statistics. The result shows that when the volume ratio of ethanol reaches 0.6, A_295_/A_260_ increases significantly, which further indicates that the ethanol at this content can cause structural changes in dsDNA.

### 3.4. Sugar Pucker Changes after Treatment with Different Inducers

In addition to the sugar–phosphate backbone structure, the sugar loop folding form is also one of the major differences between B-DNA and Z-DNA. For d(GC)_8_ in Z-DNA, the G base is in cis conformation with C3′-endo, while the C base is in inverse conformation with C2′-endo [1,67]. Among them, C3′-endo is the main structural difference between Z-DNA and B-DNA. The characteristic signals of C3′-endo in the FTIR spectrum include the peaks at 1411 cm^−1^, 1355 cm^−1^, and 1320 cm^−1^, which are mainly concentrated in 1450 cm^−1^−1150 cm^−1^ (Figure 5a−c).

Figure 5b–d show that C3′-endo is closely related to Z-DNA as its content increased with the increase of the content of the Z-DNA inducer, and its change trend is the same as that of A_1125_/A_970_ and A_925_/A_970_, indicating that Z-DNA and B-DNA can also be distinguished by analyzing the infrared spectral features that are related to sugar loop folding.

### 3.5. Z-DNA Transition in GC-Rich DNA

B-Z transition may occur in pure GC-DNA and GC-rich DNA. In reality, GC-rich DNA is more prevalent than pure GC-DNA in a genome, so we also further examined GC-rich DNA with the involvement of other bases in the DNA sequence [5]. For this purpose, we also designed d(ATATGCGCGCGCGCGCGCGCATAT), where the actual Z-DNA proportion is less than 66.7% because of the presence of a BZ junction [67,68]. As expected, FTIR spectroscopy could sensitively detect the Z-DNA transformation, and the results of this experiment just verified the B-Z transformation trend. It can be seen in Figure 6 that the characteristic spectral feature of Z-DNA remained, which essentially resembles that of the pure GC-DNA, having the Z-DNA bands such as 925 cm^−1^, 1016 cm^−1^, 1125 cm^−1^, 1320 cm^−1^, etc. It can be found that the GC proportion is directly associated with the intensities of the characteristic bands of Z-DNA. In addition, we also inspected the IR spectrum of the negative control d(AT)_8_ sequence (Appendix A), which showed no obvious characteristic peaks of Z-DNA in all the three induction solutions, confirming the usefulness of FTIR spectroscopy in the identification of Z-DNA conformation and the analysis of B-Z DNA transitions in solutions.

## 4. Discussion

First of all, to establish a reliable quantitative analytical method via FTIR spectroscopy, sufficiently stable Z-DNA should be available. In the literature, for the induction of stable Z-DNA, the following interactions or mechanisms may be involved: (1) electrostatic interactions between cations and DNA phosphate residues, such as NaCl, MgCl_2_, spermine-functionalized C-dots, etc. [21,41,69,70]; (2) reducing the degree of hydration which may promote the formation of Z-DNA conformations with a low degree of hydration, such as non-oriented gel, ethanol, etc. [37,65]; (3) protein Zα domain binding to DNA, such as ADAR1, E3L, etc. [13,71,72]; and (4) construction of negative superhelix structures, such as circular Z-DNA [23,73]. Accordingly, in this work, we applied NaCl, MgCl_2_, and ethanol to induce Z-DNA, which actually are the typical Z-DNA induction methods in the literature [20,41,42,43].

Our results showed that, compared with CD methods, FTIR spectroscopy could not only detect the structural changes of dsDNA but also provides more precise information on the DNA structure, because it can sensitively reflect the change process of B-Z transition, or even transition to other forms of DNA. For example, in Figure 2, multiple characteristic peaks of Z-DNA were clearly detected by FTIR spectroscopy in the 3 M NaCl solution. However, Z-DNA could not readily be identified from the CD spectrum when the NaCl concentration was below 3 M. In addition, Figure 3 shows that the characteristic peak of Z-DNA could be detected with the 0.5 M MgCl_2_ treatment, while the CD spectrum at the same concentration could only simply show some difference at the peak value of 285 nm, which was not enough to distinguish the Z-DNA content. This means that FTIR is more sensitive than CD in the identification of Z-DNA. The same is true for the analysis of ethanol, a noncationic inducer. As shown in Figure 4, the characteristic peak of Z-DNA appeared when the ethanol volume ratio was 0.4, while the CD spectrum showed the structural features of Z-DNA only when the ethanol volume ratio was 0.6. When the volume ratio of ethanol increased to 0.65 and 0.7, the FTIR spectra of dsDNA showed many new changes, while the CD spectra showed fewer changes.

By analyzing the second derivative FTIR spectra of B-DNA, it can be recognized clearly that dsDNA of d(GC)_8_ sequence has a characteristic peak of 1068 cm^−1^ (Figure 2b). Existing studies refer to a DNA structure as B*-DNA, which is characteristic of GC sequence nucleic acids [30,31,37]. When the concentration of NaCl reached 3 M, the concentration of MgCl_2_ reached 1.5 M, and the volume ratio of ethanol reached 0.6, this 1068 cm^−1^ peak disappeared. With the disappearance of the 1068 cm^−1^ peak, two new peaks of 1077 cm^−1^ and 1066 cm^−1^ appeared (Figure 2b, Figure 3b and Figure 4b). This result is consistent with the study of Z-DNA with the infrared spectrum by Christine Rauch et al. [38].

In addition to the characteristic peaks of Z-DNA, such as 1320 cm^−1^, 1125 cm^−1^, and 925 cm^−1^, the intensity of the 1016 cm^−1^ furanose peak also increased with the increase of inducer content. Different from Z-DNA characteristic peaks, furanose also exists in B-DNA, but is significantly enhanced in Z-DNA [47,48,49]. In this study, the area ratio (Appendix A) and intensity ratio (Appendix A) of the 1016 cm^−1^ peak to the 970 cm^−1^ peak in the process of Z-DNA transition showed a consistent variation trend with that of Z-DNA content, suggesting that the 1016 cm^−1^ peak could also be used as a marker for Z-DNA transition.

Both NaCl and MgCl_2_ may induce dsDNA to transform into Z-DNA through electrostatic interaction. Due to the large dielectric constant of water, the combination of cation and phosphate residues will be hindered [66]. Therefore, a high salt concentration is often required for the induction of Z-DNA. Na^+^ and Mg^2+^ carry different amounts of charge, resulting in different ion concentrations. In addition, when the concentration of MgCl_2_ is too high, the electrostatic interaction between MgCl_2_ and nucleic acid exceeds the requirement of Z-DNA structure transformation, which would then cause the DNA structure to change further.

Z-DNA is a conformation with a low degree of hydration. Ethanol solution can increase the degree of environmental hydration to induce the formation of Z-DNA. When the proportion of ethanol in the solution is too high, it will also go beyond the hydration range of Z-DNA. Compared with other inducers, ethanol induced Z-DNA showed significant difference in the FTIR range of 1100 cm^−1^−1000 cm^−1^, indicating that the induction mechanism of ethanol is different from that of the cation. In addition, ethanol as a polar organic solvent has a much lower permittivity than water, thus allowing the neutralization of phosphate residues by cations in solution [74]. Johan H. van de Sande et al. also demonstrated the induction of Z-DNA using ethanol with a volume ratio of 0.2 and 1 mM MgCl_2_ [20].

No matter what kinds of inducers were used in this study, all the above results showed that the use of FTIR spectroscopy could identify Z-DNA readily, monitor the B-Z DNA transition quickly, and evaluate the Z-DNA content quantitatively. Based on the results of this work, we will further continue to study the DNA B-Z transition under more complicated cases such as under the physiological conditions where Z-DNA may be induced with proteins such as ADAR1.

## 5. Conclusions

This work has established a convenient analytical assessment method based on FTIR spectroscopy for the evaluation of the B-Z transformation of DNA under different induction circumstances. In the experiment, dsDNA (d(GC)_8_) was treated with different concentrations of NaCl, MgCl_2_, and ethanol, with different volume ratios, and the Z-DNA transition processes under different conditions were observed and compared using FTIR spectroscopy. It was found that with the increase of NaCl, MgCl_2_, or ethanol concentration in the induction solution, the characteristic Z-DNA peaks of 1320 cm^−1^, 1125 cm^−1^, and 925 cm^−1^ in the infrared spectra changed during the B-Z DNA transition, and the FTIR intensity ratios of A_1320_/A_970_, A_1125_/A_970_, and A_925_/A_970_ could be used to assess the transition quantitatively. Due to the comparative analysis, we found that MgCl_2_ had the strongest effect, while NaCl had the weakest effect on the B-Z DNA transition. Compared with other detection approaches such as CD spectroscopy, FTIR spectroscopy showed the advantages of providing not only more information on multiple structural changes of DNA, but also a more sensitive detection of Z-DNA variation. As such, this study has systematically explored the changes of FTIR spectra in the process of Z-DNA transformation, which may provide a more reliable reference for the study of Z-DNA in biological systems in the future.

## Figures and Tables

**Figure 1 biomolecules-13-00964-f001:**
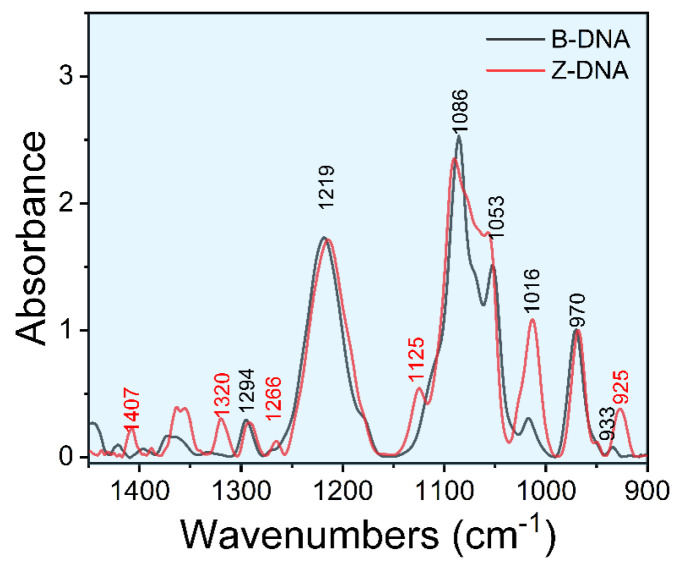
Fourier transform infrared spectrum of d(GC)_8_ dsDNA.

**Figure 2 biomolecules-13-00964-f002:**
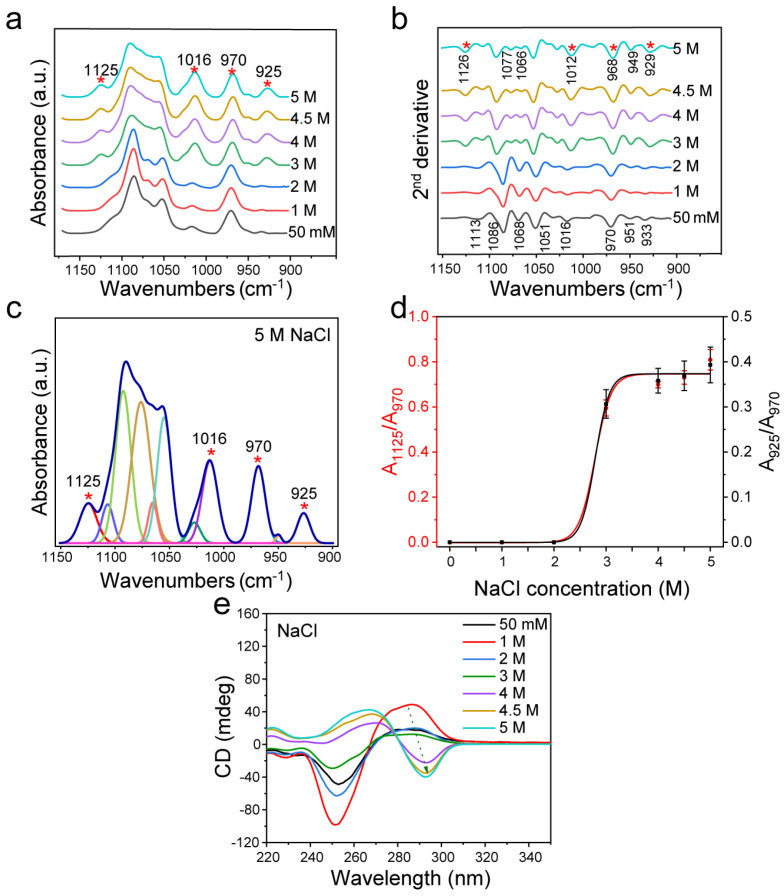
FTIR spectroscopy analysis of d(GC)_8_ treated with different concentrations of NaCl. (**a**) d(GC)_8_ infrared spectra treated with different concentrations of NaCl. (**b**) Second derivative spectra of d(GC)_8_. (**c**) The curve fitting result under 5 M NaCl (1150 cm^−1^−900 cm^−1^). (**d**) The ratios of the area of Z-DNA characteristic peaks 925 cm^−1^ and 1125 cm^−1^ to the area of internal reference 970 cm^−1^ peak (the value is from the fitting result of the characteristic peak, see the Appendix A). (**e**) CD was used to detect d(GC)_8_ chirality after treatment with different concentrations of NaCl.

**Figure 3 biomolecules-13-00964-f003:**
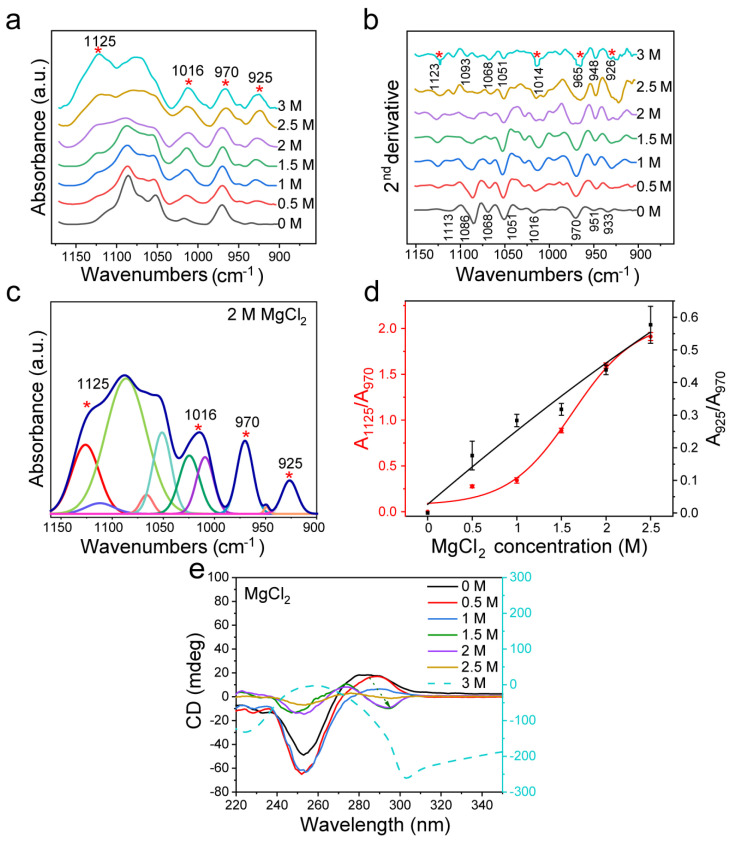
FTIR detection of d(GC)_8_ after treatment with different concentrations of MgCl_2_. (**a**) Infrared spectra of d(GC)_8_ treated with different concentrations of MgCl_2_. (**b**) Second derivative of the infrared spectrum of d(GC)_8_ treated with different concentrations of MgCl_2_. (**c**) The curve fitting result under 2 M MgCl_2_ treatment (1150 cm^−1^−900 cm^−1^). (**d**) The ratios of the area of the Z-DNA characteristic peaks 925 cm^−1^ and 1125 cm^−1^ to the area of the internal reference 970 cm^−1^ peak (the value is derived from the fitting results of characteristic peaks, see Appendix A). (**e**) CD was used to confirm the transformation effect of d(GC)_8_ treated with different concentrations of MgCl_2_.

**Figure 4 biomolecules-13-00964-f004:**
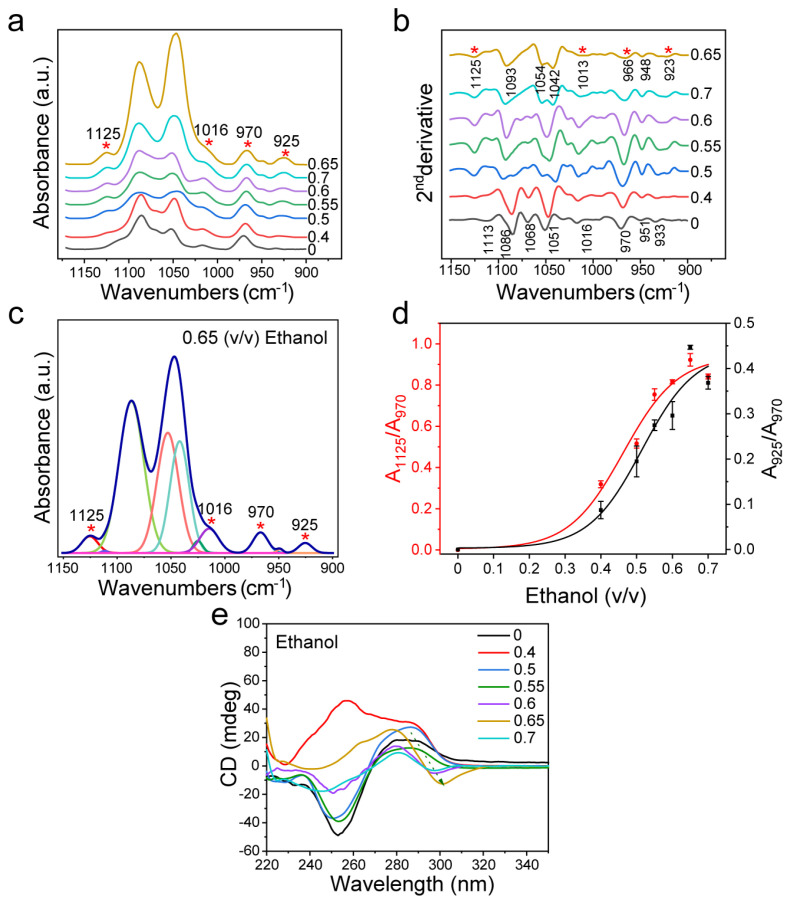
FTIR spectroscopy analysis of d(GC)_8_ treated with different volume ratios of ethanol solutions. (**a**) d(GC)_8_ infrared spectrum after ethanol treatment at different volume ratios. (**b**) The second derivative of IR spectrum of d(GC)_8_ treated with different volume ratios of ethanol. (**c**) The curve fitting result at ethanol volume ratio 0.65 (1150 cm^−1^−900 cm^−1^). (**d**) The ratio of the area of Z-DNA characteristic peaks 925 cm^−1^ and 1125 cm^−1^ to the area of internal reference 970 cm^−1^ peak (the value is derived from the fitting results of characteristic peaks, see Appendix A). (**e**) The detection result of d(GC)_8_ CD after ethanol treatment at different volume ratios.

**Figure 5 biomolecules-13-00964-f005:**
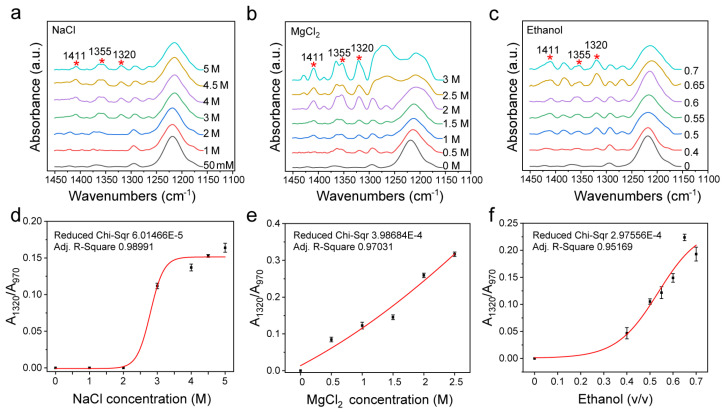
Changes of infrared spectra of 1450 cm^−1^−1150 cm^−1^ and area ratio of C3′-endo to internal reference under different inducers. (**a**) The infrared spectrum under different concentrations of NaCl. (**b**) The infrared spectrum under different concentrations of MgCl_2_. (**c**) The infrared spectrum of d(GC)_8_ treated with different volume ratios of ethanol solution. (**d**) Changes of A_1320_/A_970_ ratio under different concentrations of NaCl. (**e**) Changes of A_1320_/A_970_ ratio under different concentrations of MgCl_2_. (**f**) Changes of A_1320_/A_970_ ratio under different volume ratios of ethanol.

**Figure 6 biomolecules-13-00964-f006:**
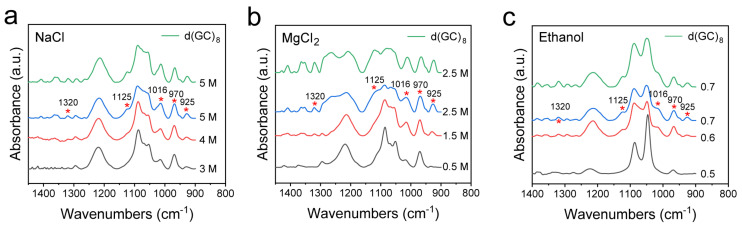
Comparison of Z-DNA infrared spectra between pure GC-DNA d(GC)_8_ and GC-rich DNA induced by different conditions. (**a**) FTIR spectra of d(ATATGCGCGCGCGCGCGCGCATAT) dsDNA treated with different concentrations of NaCl, and FTIR spectra of d(GC)_8_ treated with 5 M NaCl; (**b**) FTIR spectra of d(ATATGCGCGCGCGCGCGCGCATAT) dsDNA treated with different concentrations of MgCl_2_ and FTIR spectra of d(GC)_8_ treated with 2.5 M MgCl_2_; (**c**) FTIR spectra of d(ATATGCGCGCGCGCGCGCGCATAT) dsDNA treated with different volume ratios of ethanol, and FTIR spectra of d(GC)_8_ treated with 0.7 volume ratio of ethanol.

**Table 1 biomolecules-13-00964-t001:** Assignment of the relevant vibrational bands used in the second derivative.

Wavenumber (cm^−1^)	Vibrational Mode Assignment and Main Contribution
1408~1413	C3′-endo deoxyribose in Z-form helices [47,48,49]
1352~1357	Purine in syn conformation (sugars: C3′-endo) [47,48,49]
~1320	G in syn conformation (sugars: C3′-endo) [47,48,49,50]
1213~1216	Main Z-form marker, antisymmetric ${PO}_{2}^{-}$ stretch [47,49,51]
~1123	Z-form [47]
~1086	Symmetric stretching vibrations of phosphate moieties in nucleic acids [52,53,54]
1044~1069	CO stretch of backbone, strongly enhanced in Z-form DNA [47,48,55]
1010~1020	Furanose vibration, strongly enhanced in Z-form DNA [47,48]
~970	B-form: singlet at 970 (CC str of the backbone) [55]
~951	Z-form [48]
~935	B-DNA/Deoxyribose (ν(CC)ring) [56]
924~930	Z-form (ν(OPO)backbone) [47,56]

## Data Availability

The data presented in this study are available in this article and its Appendix A.

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
