# Peer review of "Assessing B-Z DNA Transitions in Solutions via Infrared Spectroscopy"

_biomolecules, 2023, doi:10.3390/biom13060964_

Round 1
Reviewer 1 Report
The authors present a study of the transition from the right handed B conformation to the left-handed Z conformation of DNA oligonucleotides that are rich in GC content. Specifically, they use FTIR as the method to evaluate the transition and come to the conclusion that FTIR spectroscopy offers advantages over other physical techniques used to study this transition. The FTIR data have much more information content than methods such as circular dichroism spectroscopy.
In this Reviewer's opinion, the manuscript needs extensive revision before it can be published. It would be best if a native speaker of English could help the authors with the manuscript.
The authors write many unusual things, far too many to cite in one review. For example, "As shown by the CD spectra in Figure 2e, dsDNA was obviously transformed to Z-128 DNA after treatment with NaCl with concentration greater than 4 M," This has been shown in several previous publications, some of them dating to more than 40 years ago. Why is it here without a citation?
It is not clear to this Reviewer why the authors chose to fit the data in Figures to the equation Y=A2+(A1-A2)/(1+exp((x-x0)/dx)).
Without going thru an extensive literature search, this Reviewer was surprised that an FTIR study of the B-Z transition hadn't been published previously. However, this Reviewer is not aware of any and indeed, this might be the first. However, it is important to point out that this Reviewer found the following citation by doing a simple Google search:
- Characterization of Z-DNA by Infrared Spectroscopy
- Fengqiu Zhang & Qing Huang
- Protocol
- First Online: 10 March 2023
- 208 Accesses
- Part of the Methods in Molecular Biology book series (MIMB,volume 2651)
As the paper cited above is by the same authors, it is important that they explain how the present manuscript differs from that of their previous work.
Conclusion, extensive revision.
The English needs extensive revision.
Reviewer 2 Report
In this manuscript, Zhang, Huang and coworkers reported on the use of FT-IR spectroscopy to monitor the B-Z transition of GC-rich DNA. The authors used NaCl, MgCl2, and ethanol to induce a transition and also recorded CD spectra to benchmark the resultant transformation. The results from this work appears to further support FT-IR as a viable spectroscopy for detecting Z-DNA formation in aqueous solution.
The IR data suggest that, among the three inducing agents used, NaCl is the most effective as well as the most useful in terms of spectral changes in the IR spectra e.g., induced peaks at 1016 and 925 cm-1. The spectral changes induced by MgCl2 and ethanol are more ambiguous. Indeed, the spectra with ethanol do not appear to change monotonically as a function of inducer concentration e.g., Figure 4a. Overall, the strongest positive evidence for Z-DNA conversion in the data is the inversion in the CD signal around 290 nm as a function of inducer concentration. The CD signal towards the shorter wavelengths are likely suspect given the extremely high concentrations of Cl- and ethanol reached in the samples.
My major suggestion is for the authors to investigate the IR spectrum of a negative control sequence that does not undergo a B-Z transition and compare the resultant spectral changes. Such a control would be definitive evidence for the utility of the IR spectrum as a quantitative marker of Z-DNA formation rather than the ATAT(GC)8ATAT sequence tested in Section 2.5.
In terms of presentation, the transition data should be plotted with concentration on a numerically scaled abscissa. The current presentation in which wildly unequal concentration increments are spaced evenly obscures the reader’s interpretation of the transition.
Language is generally okay but can benefit from some editing.
Reviewer 3 Report
Assessing B-Z DNA transition in solutions via infrared spectroscopy
Mengmeng Duan , Yalin Li , Fengqiu Zhang , and Qing Huang
The paper analyzes the B-Z transition under different ion and alcohol conditions using and Fourier transform infrared (FTIR) spectroscopy in the region of 1450 cm-1-900 cm-1 region, concluding that ratios of A1320/A970, 23 A1125/A970, and A925/A970 signals measures the transition to Z-DNA. As referenced in Table 1 this approach has been extensively studied.
This approach has also been used to study Z-DNA in foxed cells e.g. 10.1002/cyto.a.22585 and 10.1021/acs.analchem.0c02432 where background signals due to other cell constituents are considered.
The study of Duan et al therefore does not cover any new ground, nor does it evaluate their method under physiological conditions using proteins such as Z22 antibody or Zσ to induce Z-DNA.
The hope of biologists is that FTIR can be developed to follow Z-DNA and Z-RNA formation on live cells. This paper does not advance that cause.
The quality of the English is good
Reviewer 4 Report
The article “Assessing B-Z DNA transition in solutions via infrared spectroscopy” by Duan et al. describes the practical analytical approach for B-Z transformation of DNA using FTIR spectroscopy. This topic is very interesting and also has useful contributions to the study of Z-DNA, which is suitable for publication in this journal. However, there is one point that need to be improved to make it a better paper. I believe that the manuscript can be accepted after comment below is addressed.
Comment:
The process of obtaining the fitting results should be described in detail in the Supporting Information section.
Furthermore, additional explanations should be provided in the Supporting Information section regarding the Fit parameters.
Minor comment:
Some typo in the manuscript should be corrected. Please check all typos again in your manuscript.
e.g.
In your manuscript, the authors described “(c) Peak fitting” in line 100, “(c) Spectral peak fitting” in line 173, and “(c) Peak fitting” in line 201. What is the correct wording?
Line 145, “Figure 3a and Fig 3b” should be corrected to “Figure 3a and 3b”.
Round 2
Reviewer 1 Report
Page 2. DNA, researchers have tried to induce the formation of Z-DNA in vitro under special conditions such as high ionic strength, chemical modification, binding protein, and negative superhelix [13-23]
They didn’t “try”, the did induce it.
(d) The ratio of the area of Z-DNA characteristic peaks 925 cm-1 and 1125 cm-1 to the area of internal reference 970 cm-1(the value is derived from the fitting results of characteristic peaks, see Supporting Information). The fitting function is Y=A2+(A1-A2)/(1+exp((x-x0)/dx)). Reduced Chi-Sqr for A1125/A970 is 0.0023 and Reduced Adj. R-square is 0.97969; Reduced Chi-Sqr for A925/A970 is 0.00128 and adj. R-square is 0.94769.
Why did the authors use this equation? Giving the fit parameters isn’t important if it is simply a heuristic fit of the data.
Page 8: In reality, GC-rich 313 DNA is more prevalent than pure GC-DNA in genome, so it is necessary to examine GC-rich DNA with the involvement of other bases in the DNA sequence [5].
What do the authors mean to say here?
Quoting from the Methods section:
“Firstly, the purchased ssDNA was combined into dsDNA using the base complementary pairing principle. The dry nucleic acid powder was first dissolved with buffer solution (50 mM NaCl, 5 mM Tris-HCl, pH=8.0) and then incubated at 37 °C for 30 minutes to obtain stable dsDNA [41]. NaCl, MgCl2 and ethanol were prepared in buffer solutions (50 432 mM NaCl, 5 mM Tris-HCl, pH=8.0). The prepared inducer solution was added to the dsDNA solution. The final concentration of NaCl was 1 M, 2 M, 3 M, 4 M, 4.5 M, 5 M, 434 respectively; the final concentration of MgCl2 was 0.5 M, 1 M, 1.5 M, 2 M, 2.5 M, 3 M, respectively; and the volume ratio of ethanol was 0.4, 0.5, 0.55, 0.6, 0.65, 0.7, respectively. After adding the Z-DNA inducer solution, the DNA samples were incubated at 37 °C for more than 4 hours to stabilize the structure of dsDNA.
4.3. FTIR spectral analysis
Before the FTIR spectral measurements, the dsDNA was added to a CaF2 window piece drop by drop and put into a drying oven keeping the temperature at 37 °C for about 20 minutes until the water of DNA solution evaporated on the window piece. Then, 5 μL of different inducing solutions were added on the film formed by dsDNA, and the film was quickly covered with another clean CaF2 window piece, and the connection of the window piece was sealed with sealing film to reduce the evaporation of water.”
From this description it is not clear what the final concentration of cation (salt) is in the FT-IR measurements. The DNA solutions contained salts and it is not stated how much of the DNA solution was used in each measurement. The salt from the dried DNA solutions would add to the salt in the final solution (after addition of the “inducing solutions”).
Page 10: It was found that with the increase of induced solution concentration,
Do the authors mean concentration of salt or “inducer”?
It is ok.
Reviewer 2 Report
The authors have addressed the issues I raised in the first review.
Author Response
Thanks for your comments and suggestions.Wish you all the best.
Reviewer 3 Report
I think the paper would be more interesting if you made the FTIR measurements under physiological conditions where Z-DNA was induced with proteins.
The quality of the English has improved
Author Response
Reply: Thanks for your comments and suggestions. We appreciate the suggestion that for the FTIR investigation of Z-DNA induced by proteins. In fact, in our previous work, we had induced Z-DNA under physiological conditions such as ADAR1 protein and methylation modification. But this will involve more complicated biochemical reactions which is not easy to obtain the optimal Z-DNA condition. In this work, our intention to establish the analytical method on a solid basis, with this, we will certainly to continue the DNA B-Z transition study under more complicated cases such as under the under physiological conditions where Z-DNA was induced with proteins. Again, thanks for your good suggestion and understanding.